# Impact of Job Demands and Resources on Nurses’ Burnout and Occupational Turnover Intention Towards an Age-Moderated Mediation Model for the Nursing Profession

**DOI:** 10.3390/ijerph16112011

**Published:** 2019-06-05

**Authors:** Beatrice Van der Heijden, Christine Brown Mahoney, Yingzi Xu

**Affiliations:** 1Head of Department Strategic HRM/Full Professor of Strategic HRM, Institute for Management Research, Radboud University, P.O. Box 9108, 6500 HK Nijmegen, The Netherlands; 2Faculty of Management, Science & Technology, Open University of the Netherlands, P.O. Box 2960, 6401 DL Heerlen, The Netherlands; 3Faculty of Economics and Business Administration, Ghent University, Tweekerkenstraat 2, 9000 Ghent, Belgium; 4Kingston Business School, Kingston University, Kingston-Upon-Thames, London KT2 7LB, UK; 5Business School, Hubei University, Wuhan 430062, China; 6Professor of Management, College of Business, Minnesota State University, Mankato, MN 56001, USA; christine.mahoney@mnsu.edu; 7Faculty of Business & Law, Senior Lecturer of Marketing, Auckland University of Technology, Auckland City Central 1010, New Zealand; yingzi.xu@aut.ac.nz

**Keywords:** job resources, job demands, burnout, occupational turnover intention, JD-R model, longitudinal approach, Dutch nurses, age

## Abstract

This longitudinal study among Registered Nurses has four purposes: (1) to investigate whether emotional, quantitative and physical demands, and family-work conflict have a negative impact on nurses’ perceived effort; (2) to investigate whether quality of leadership, developmental opportunities, and social support from supervisors and colleagues have a positive impact on meaning of work; (3) to investigate whether burnout from the combined impact of perceived effort and meaning of work mediates the relationship with occupational turnover intention; and (4) whether the relationships in our overall hypothesized framework are moderated by age (nurses categorized under 40 years versus ≥ 40 years old). In line with our expectations, emotional, quantitative, and physical demands, plus family-work conflict appeared to increase levels of perceived effort. Quality of leadership, developmental opportunities, and social support from supervisors and colleagues increased the meaning of work levels. In addition, increased perceived stress resulted in higher burnout levels, while increased meaning of work resulted in decreased burnout levels. Finally, higher burnout levels appeared to lead to a higher occupational turnover intention. Obviously, a nursing workforce that is in good physical and psychological condition is only conceivable when health care managers protect the employability of their nursing staff, and when there is a dual responsibility for a sustainable workforce. Additionally, thorough attention for the character of job demands and job resources according to nurses’ age category is necessary in creating meaningful management interventions.

## 1. Introduction

Today, most developed countries in the European Union and elsewhere have a shortage of active nurses, which is likely to increase as economies improve [1,2,3,4,5]. Demographic changes within the coming two decades are likely to worsen this situation. The major contributors to the shortage are a decrease in the proportion of younger individuals entering the working population, an increase in the proportion of older people in the working population, and an increase in the number of people over 64 years in the population, as a whole. Since it is the oldest members of the population who require the most care; the demand for health care services will significantly increase [4], while, unfortunately, the formerly mentioned changes in the working population decrease the supply of nurses. Additionally, the fact that younger nurses are more likely to leave the nursing profession is further worsening the shortage of nurses [6,7].

Therefore, one way of assuring a sufficient supply of nurses in the future would be to promote the retention of existing nursing staff. Employee job turnover and leaving the profession as a whole, that is to say, occupational turnover, is a growing concern to Human Resource Development (HRD) professionals as their main goal is to develop and maintain sufficient human expertise to deliver, in case of nurses, high-quality patient care. Equally as important, organizations will bear both the direct and indirect costs of increased turnover. The direct costs of turnover are advertising and recruiting costs, which include advertising costs, costs for personnel who do the recruiting and all other expenses to recruit (e.g., at college job fairs), interviewing, background checks, etc., bonuses for new hires, training and orientation costs, and costs for personnel. Indirect costs include the costs of replacement labor; such as temporary nurses who cost more per hour than staff nurses or paying overtime to staff nurses. Moreover, organizations would lose revenue if units closed due to a lack of nursing staff. All in all, managers will have to increase their productivity when nurses experience burnout, in order to make up for lost productivity, decreased quality of patient services provided, and lower productivity of new hires [8,9]. Undoubtedly, adverse psychological and physical working conditions may contribute to the nurses’ decision to leave their profession. So far, many scholars have examined job demands’ and job resources’ impact on burnout [10], and previous research has indicated that burnout, in particular, is a strong risk factor for turnover [11]. However, to the best of our knowledge, no earlier research has investigated a model with perceived effort (or stress), being predicted by job demands, and meaning of work, being predicted by job resources, simultaneously impacting burnout; and with burnout mediating their relationship with occupational turnover intention of nurses. Therefore, the focus of this study is to better understand whether job resources, through their impact on the meaning of work, may buffer or compensate for the effect of job demands. The latter, through their impact on effort/strain, are assumed to be positively associated with burnout, which, in turn, is assumed to be a predictor of nurses’ occupational turnover intention. Up to now, most research focused upon organizational turnover and less on occupational turnover [12]. As such, this study adds to calls from the previous scholarly literature by focusing on nurses’ intention to leave their profession as a whole [13]. 

During the past decades, many studies have shown that unfavorable job characteristics may have a strong relationship with job stress and burnout [14]. However, notwithstanding the increase of insight into possible antecedents of burnout, theoretical insight is still limited. Bakker, Demerouti and Euwema [15] in their excellent contribution wherein they extended the Job Demands-Resources (JD-R) model [16,17], tested whether burnout may be the result of an imbalance between job demands and resources, and whether *several* job resources may compensate for the impact of several job demands on burnout. 

The empirical work that is reported in this article aims to test the generalizability of the JD-R model for the nursing profession. In particular, a longitudinal study with a questionnaire completed twice (1-year time lag) by Registered Nurses working in hospitals (63.4%), nursing and old peoples’ homes (15.4%), and home care (21.1%) was conducted. Our final sample comprised 1,187 nurses, with 5.4% men and 94.4% women. Their mean age was 39.8 years (SD = 9.68). Nurses’ jobs are typically stressful and emotionally demanding as nurses are confronted with peoples’ needs, problems and suffering all the time, and also with serious illness and death. Burnout affects approximately 25% of nurses, and they are considered to be particularly susceptible to burnout. This ratio reaches even 64% among nurses with high affective strain, and 39% among those with high cognitive strain (see [18]) for a large-scale study using French nurses. Similar findings have been reported from many other countries: 43% in China [6], over 50% in Sweden [19], and 37% in Turkey [20]. The costs of burnout may be very high, especially when a nurse is not able to cope with the increasing workload, experiences defeats, and lack of professional success [21]. 

In case of evidence for the generalizability of the JD-R model, our research findings may be a starting-point for the development of goal-directed preventive measures within health care institutions. Buffering variables may reduce the effect of specific stressors, alter the perceptions and cognitions evoked by such stressors, moderate responses that follow the appraisal process, and/or reduce the health-damaging consequences of these responses [22]. That is to say, proof for such buffering effects implies that nurses’ well-being and even their employability (or career potential) [23,24,25] may be maintained, even when it is difficult to reduce the amount of job demands. Therefore, it is highly necessary to conduct empirical research aimed at enlarging our understanding of so-called sustainability at work, in particular of how to prevent burnout and turnover among nurses. 

Our empirical results strongly support that workplaces reducing quantitative demands on nurses will lead to the greatest decrease in occupational turnover intention. Specifically, this reduction in quantitative demands will lead to much lower perceived effort levels, lower burnout, and finally, lower probability of occupational turnover intention. Additionally, increasing job resources, that is, quality of leadership, developmental opportunities, and social support from supervisors and near colleagues will increase levels of meaning of work, lower burnout, and result in lower probability of occupational turnover intention. The higher nurses’ job demands, the higher their level of burnout, and the more likely they are to leave the nursing profession over time. Management in health care organizations can lower the probability of nurses’ turnover intention by investing in sound job resources.

## 2. Theory

### 2.1. The Job Demands-Resources (JD-R) Model

The JD-R model is built upon two underlying psychological processes that play a role in the development of job strain and motivation [26]. The first one comprises a so-called health-impairment process, a situation wherein a too high amount of job demands (i.e., job pressure, such as a high amount of emotional demands and work-home conflicts for the nursing staff) exhausts employees’ mental and physical resources and may therefore lead to exhaustion, health problems, and eventually premature leave from their profession. The second underlying process is motivational in nature and comprises that job resources (i.e., general resources and job recognition, such as possibilities for development and influence at work) have either intrinsic (because they foster growth, learning and development) or extrinsic (because they are instrumental in achieving work goals) motivational potential, and lead to positive work outcomes, such as work engagement [27], and high job performance [26]. As such, job resources are necessary in order to deal with job demands, but they are also rewarding in themselves, by fulfilling basic human needs [28], such as the needs for autonomy, belongingness, and competence. Both of these processes, the one created by job demands and the one created by job resources, occur simultaneously, not sequentially [29].

### 2.2. Towards a Nursing Sector-Specific Design of the Job Demands-Resources Model

In this study, we will empirically investigate the central notion whether particularly the combination of high job demands and low job resources is predictive of burnout in the nursing sector. The proposed simultaneous effect is tested using *four specific job demands* (emotional demands, quantitative demands, physical demands, and family-work conflict) and *four specific job resources* (quality of leadership, developmental opportunities, social support from supervisor, and social support from near colleagues). The JD-R model emphasizes that the selection of concrete demands and resources for scholarly work is dependent on the occupational sector wherein specific research is conducted [16]. Based upon earlier research within the nursing sector [4], and following the theoretical framework by Bakker, Demerouti, and Euwema [15], we concluded that these categories of job demands and job resources [30,31,32,33,34] were crucial in the light of work-related outcomes, in our case burnout, and, subsequently, occupational turnover intention. 

Previous research on Leader-Member eXchange (LMX) indicates the importance of the relationship between supervisor and subordinate, or *the quality of leadership*, in the light of organizational outcomes, such as well-being [35]. We assume, in line with Bakker, Demerouti, and Euwema [15] that high-quality leadership may alleviate the negative effects of job demands on burnout, because supervisors’ appreciation and support put demands in another perspective. 

*Developmental opportunities* are, obviously, also highly important as a stress-buffering factor within the nursing sector. A job with a high value as a nutrient for further professional development, and wherein one is enabled to learn new knowledge and skills, enhances one’s employability [36,37,38]. Tasks, responsibilities, and duties that are sufficiently challenging are one of the strongest motivators that a work environment can offer. That is, it is the best preventive medicine for becoming obsolete or becoming an ineffective plateauee [39]. Within the nursing sector, jobs should be rich in resources, tools and learning materials, and they should offer ample opportunities for social interaction and collaboration. Tasks should be varied and to some degree unpredictable to enable nurses to improve their performance. We posit that developmental opportunities may buffer the impact of job demands upon burnout, as increased performance and a growth in capabilities reduce the tension that is experienced by the employee. 

Moreover, in line with De Jonge, Mulder, and Nijhuis [40] and Houkes, Janssen, De Jonge, and Nijhuis [41] who advocated the examination of more specific predictions regarding work characteristics and work reactions, this contribution focuses upon the potential buffering effect of *social support from different parties* (in our case, immediate supervisor and close colleagues) upon burnout, and, in turn, occupational turnover intention. The power of buffering variables was extensively explained by Van der Doef and Maes [42] who dealt with the protective effect of social support. Similarly, Rhoades and Eisenberger [43] have concluded that perceptions of supportive HR practices, such as organizational rewards (e.g., recognition, opportunity for advancement), procedural justice (e.g., communication, decision-making), and supervisory support (e.g., concern for employees’ well-being) led to perceived organizational support (e.g., organizational concern), which, in its turn, led to affective organizational commitment (e.g., sense of belonging or integration, and attachment). Integration of employees is supposed to be achieved through both formal and informal means. Formal experiences are deliberately planned interactive events (e.g., formal communication lines, policies, and meetings) while informal experiences would tend to be more spontaneous opportunities to interact. As the opportunities for interactions, and information and feedback exchange overlap, and often involve the same people (e.g., supervisors and peers), formal and informal dimensions are connected and interrelated [44]. 

Analogously, Estryn-Behar [45], in her exemplary review on cognitive (e.g., interruptions in tasks, need of frequent reorganization of daily work program, and overwork) and affective strain (e.g., adequacy between training and actual tasks, time to talk to patients and answer to their questions, satisfaction with job climate, and interest of the job) in health care, stressed that nurses’ ability to cope with stress depends upon the extent of their support network and their possibility to discuss and improve patient’s quality of life [18]. Moreover, interpersonal relationships appear to be important predictors of job satisfaction [46], and, consequently, related to absenteeism, expression of grievances, and turnover [47,48].

Therefore, we assume that nurses will show less burnout if they experience high levels of support from their direct supervisor, and from close colleagues. The so-called stress-buffering hypothesis states that social support protects employees from pathological consequences of stressful experiences [49]. Research outcomes pertaining to the health care sector, in particular, have indeed indicated the positive impact of counselling and interactions between staff members, and between nurses and physicians [18,50]. 

All in all, building upon the JD-R model, we argue that job demands are costly [51] as workers, in our case nurses, who are confronted with high job demands are necessitated to spend time and energy to engage in performance-protection strategies by investing psychological and physiological resources [32]. Following the Conservation of Resources (COR) theory [52], it is this depletion of resources, due to coping with high demands, that evokes stress [32]. Specifically, COR theory is built upon the concept of resources, which refer to things that people value, such as objects, but also conditions (for instance the quality of the relationship with one’s supervisor, and developmental opportunities at work), personal characteristics (i.e., one’s age) and energies, and social support from one’s supervisor and close colleagues [53]. We use COR theory to provide an overarching framework for understanding occupational turnover intention [54]. In particular, drawing from the notion that people seek resources to fulfill their goals [52], nurses will remain in their health care organization in case it provides the resources they need. When they experience loss of resources and/or increasing demands, they are likely to perceive that the achievement of their goals and their well-being is at risk, and their occupational turnover intention may increase.

Based on the theoretical outline given so far, we have formulated the following hypotheses:

**Hypothesis** **1:**
*Perceived effort and burnout mediate the relationship between job demands and occupational turnover intention.*


**Hypothesis** **2:**
*Meaning of work and burnout mediate the relationship between job resources and occupational turnover intention.*


**Hypothesis** **3:**
*Burnout mediates the relationship between perceived effort and occupational turnover intention.*


**Hypothesis** **4:**
*Burnout mediates the relationship between meaning of work and occupational turnover intention.*


In addition, we argue that it is of utmost important to gain more insight into the role of age in the so-called *Nursing Sector-Specific JD-R model* that is empirically investigated in this contribution. Unfortunately, few researchers have studied differences in model relationships for distinguished age groups [55]. However, differences in career outcomes, such as occupational turnover intention, depending upon employee’s age, are plausible considering the prevalence of age-related stereotyping [56], resulting in differential treatment for older versus younger workers, and increased Person-Environment (P-E) fit for older workers [57], resulting into a more clearly defined self-concept with age. Career choices, including occupational turnover, comprise processes of matching one’s self-concept with images of the occupational world [58]. In a similar vein, Wright and Hamilton’s [57] ‘job change’ hypothesis states that due to experience, seniority and skills, it is likely that older workers will have obtained a relatively better P-E fit [59]. Therefore, we argue that older workers have a lower occupational turnover intention. 

Numerous studies [60,61,62,63,64,65,66] already suggested specific reasons for why older nurses are less likely to leave their jobs: they have greater firm-specific human capital, their pay is higher, their position allows more autonomy, power, or status, and they participate more in decision-making. These factors lead to increased job satisfaction, greater perception of distributive justice, and increased organizational commitment, which, in turn, decrease the desire for turnover. Older nurses are also more likely to have more close friends in their workplace, increased ties to community and local organizations, and more obligations to kin; altogether, these factors result in higher psychological costs of leaving the organization. 

All in all, earlier scholarly work indeed provides substantial support that a negative relationship exists between age and turnover, starting with Price’s [67] seminal review and study of turnover. In particular, Price [66] tested an explanatory model of turnover that supported the assumption of a negative relationship between age and turnover for registered nurses. This negative relationship holds in studies of nurses from many countries [68,69,70,71]. 

Building upon the theoretical outline given above and, as regards the age distribution, in particular on Van der Heijden [12], also Finkelstein [72] (p. 100) on the Age Discrimination in Employment act (ADEA), we categorize nurses into younger (under 40 years) versus older (≥ 40 years old) ones, and have formulated the following hypotheses: 

**Hypothesis** **5:**
*An increase in age will result in decreased intention of occupational turnover.*


**Hypothesis** **6:**
*The determinants of turnover are not identical for those under the age of 40 and those aged 40 and over.*


To conclude, in this scholarly work we aim to test and refine the JD-R model among a considerable sample of nurses in the Netherlands. Nursing, like any other profession, has its own specific risk factors that are associated with job stress. The central notion of the JD-R model [17,26] is that burnout is the result of an imbalance between job demands and resources, and that several job resources may compensate for the influence of several job demands upon burnout. Specifically, job demands, although not necessarily negative [73] may result in stress when meeting those demands requires a too high level of effort for which the employee is not adequately trained or supported for to perform well. Job resources, on the other hand, are valued as being important means to either manage high levels of job demands or to protect valued resources. 

## 3. Methods

### 3.1. Sample and Procedure

To select the Dutch health care institutions, for each region (north, south, east, west, and middle), a careful sampling strategy across hospitals, nursing and old peoples’ homes, and home care institutions was conducted. The selection was based upon information regarding the distribution of the Dutch nursing population that was obtained from the Internet, from the Chamber of Commerce, and from national federations of health care institutions. Although we tried for a representative sample, convenient sampling was used as well. Due to previously performed large surveys (e.g., [74]), many health care institutions’ management boards, especially in the western part of the Netherlands, indicated that their employees showed research fatigue and were reluctant to participate. Moreover, following economic drawbacks, many institutions were in the middle of a fusion and/or reorganization implying that the management team decided not to participate to scientific studies in order to prevent unnecessary stress for their workforce. 

After contacting the network of Cooperating Top Clinical Hospitals in the Netherlands, another three hospitals decided to participate in our study. In total, 27 health care institutions (nine hospitals, thirteen nursing and old peoples’ homes, and five home care institutions) decided to participate in our study. In each participating institution, a thorough discussion with a representative from the personnel department took place. We carefully explained the criteria for participation, and we composed samples of nurses. The confidentiality and anonymity of the data were emphasized. In order to facilitate data gathering, in each participating institution, a contact person was pointed out. Most health care institutions distributed the questionnaires themselves among the participants. For two institutions, we have sent the questionnaires to the home addresses. Two other institutions took care of sending the questionnaires to the home addresses of the nurses. In the remaining institutions, our contact persons made sure that the questionnaires were handed out at work meetings, or distributed by means of the nurses’ mailboxes at the health care institution. The contact persons have been approached several times by phone and in person to alert them to remind the respondents to fill out and to return the questionnaire. 

Our research design is longitudinal and comprises two measurements. All those nurses who participated in the first survey (Time 1) received an additional questionnaire, the so-called follow-up survey, twelve months after they filled out the first one (Time 2). A total of 1,187 nurses filled out and returned both the first and second wave questionnaires. In general, the response rate of nurses in home care appeared to be lowest. The overall response rate for the first measurement at Time 1 was 43.6%; for the Time 2 measurement it was 29.5%. For the second measurement, the sample consisted of 753 (63.4%) Registered Nurses working in hospitals, 183 (15.4%) nurses working in nursing and old peoples’ homes, and 251 (21.1%) nurses in home care institutions. The sample included 66 men (5.4%) and 1,121 women (94.4%). The mean age was 39.8 years (SD = 9.68). The average number of years of working experience in the nursing profession was 13.6 years (SD = 8.57). 

### 3.2. Measures

*Job demands.* Four job demands’ factors were included in the present study. Emotional demands were measured using De Jonge et al.’s [40] four-item scale developed specifically for health care professions. The scale has five response categories ranging from (1) ‘never’ to (5) ‘always’ and measures how often the nurses were confronted with ‘death’, ‘illness or any other human suffering’, ‘aggressive patients’, and ‘troublesome patients’ in their work. The internal consistency reliability estimate using Cronbach’s alpha was 0.70.

Quantitative demands were assessed using four items of the quantitative demand scale of the Copenhagen Psychosocial Questionnaire (COPSOQ) [75] and one additional item was added by the NEXT-Study Group. COPSOQ items are: ‘How often do you lack time to complete all your work tasks?’, ‘Can you pause in your work whenever you want?’, ‘Do you have to work very fast?’ and ‘Is your workload unevenly distributed so that things pile up?’ The additional item was: ‘Do you have enough time to talk to patients?’ Responses used a five-point rating scale (1 = hardly ever, 5 = always). The internal consistency reliability estimate using Cronbach’s alpha was 0.70.

Physical demands were assessed by using a newly developed scale entitled ‘lifting and bending’. The scale was designed to quantify the physical demands of the nursing profession and consisted of eight items [4]. The eight items are: ‘bedding and positioning patients’, ‘transferring or carrying patients’, ‘lifting patients in bed without aid’, ‘mobilizing patients’, ‘clothing patients’, ‘helping with feeding’, ‘making beds’, ‘pushing patient’s beds, food trolleys, or laundry trolleys’. Response categories are: (1) ‘0–1 times a day’, (2) ‘2–5 times a day’, (3) ‘6–10 times a day’ (4) ‘> 10 times a day’. The index for lifting was composed of a score for the first four items, added and divided by four and multiplied by 25. The index for bending was composed of a score for the remaining four items, added and divided by four and multiplied by 25. These indices were summed and divided by 20 to standardize them to a scale similar to other variables in the model. The internal consistency reliability estimate using Cronbach’s alpha was 0.87. 

Family-work conflict was measured using a scale developed by Netemeyer, Boles and McMurrian [76] and contains five items that measure home-to-work interference. A sample item is: ‘The demands of my family or spouse/partner work interfere with work-related activities.’ A five-point rating scale was used (1 = completely disagree, and 5 = completely agree). Cronbach’s alpha was 0.85 for this scale.

*Job resources.* Four job resources were included in the questionnaire. The nurses’ perception of the Quality of leadership was assessed using a four-item scale [75]. Items were designed to gather information on the superior’s engagement in supportive leadership activities aimed at providing role clarity, development opportunities, predictability, and a positive work climate. One example item is: ‘To what extent would you say that your immediate supervisor makes sure that the individual member of staff has good development opportunities?’ Responses were made on five-point rating scales (1 = to a very small extent, and 5 = to a large extent). The Cronbach’s alpha was 0.87.

Developmental opportunities as perceived by the nurses was measured using the COPSOQ [75] that contains four items. An example item is: ’Does your work require you to take the initiative?’ The scale ranges from 1 (low possibilities for development) to 5 (high possibilities for development). The internal consistency reliability estimate was 0.75.

Social support from one’s immediate supervisor was measured using a four-item scale developed by Van der Heijden [37,77]. An example item is: ‘Does your immediate supervisor regularly give you supportive advice?’ Respondents could indicate their answers on six-point Likert scales (1 = never, and 6 = very often). The internal consistency reliability estimate (Cronbach’s alpha) was 0.84 for supervisory support. 

Social support from one’s near colleagues was measured by exactly the same four items, with ’close colleagues’ substituted for ’immediate supervisor’ in the item statement (derived from Van der Heijden, [37,77]). The internal consistency reliability estimate (Cronbach’s alpha) was 0.77 for colleague support.

Perceived effort (or stress), being the first mediator in our hypothesized model, was measured using six items from the effort-reward imbalance model [78]: ‘I am under constant time pressure due to the heavy work load’, ‘I have many interruptions and disturbances in my job’, ‘I have much responsibility in my job’, ‘I am often pressured to work overtime’, ‘My job is physically demanding’, and ‘Over the past few years, my job has become more and more demanding’ [78]. Responses were given on a four-point Likert scale, ranging from 1 = no distress at all to 4 = very much distress. The internal consistency reliability estimate was 0.85. 

*Meaning of work,* being the second mediator, was measured with the COPSOQ [75] meaning of work scale, which includes the perception of motivation. Items are: ’Is your work meaningful?’, ’Do you feel that the work you do is important?’ and ’Do you feel motivated and involved in your work?’ The scale ranges from 1 (to a very small extent) to 5 (to a very large extent). The internal consistency reliability estimate was 0.81.

*Burnout* was assessed using the six-item scale from the Copenhagen Burnout Inventory (CBI) [79]. Respondents were provided with a five-point scale, which ranged from (1) ’never/almost never’ to (5) ’almost every day’, in order to indicate how frequently they experienced the following: ’feel tired’, ’are physically exhausted’, ’are emotionally exhausted’, ’think - I can’t take it anymore’, ’feel worn out’, ’feel weak and susceptible to illness’. The internal consistency reliability estimate at Time 1 was 0.83. 

*Occupational turnover intention* was measured with Hasselhorn, Tackenberg, and Mueller’s [4] three-item scale. A sample item is: ‘How often during the past year have you thought about giving up nursing completely?’ Responses were given on a five-point rating scale ranging from: (1) never, to (5) every day. At Time 2, Cronbach’s alpha was 0.85.

*Control variables* used in the model were gender, and tenure in the profession. Other control variables that could be expected to confound relationships between predictors and intention to leave nursing were included in preliminary analyses, yet, appeared to have no significant impact (e.g., hours worked per week, type of health care institution, years of education). Therefore, to facilitate model estimation and to increase the power of the statistical testing, they were excluded from all further analyses. 

### 3.3. Statistical Analyses

Analyses were done with Structural Equation Modelling (SEM) using maximum likelihood estimation within the AMOS software package, Version 25.00 (IBM SPSS, Chicago, IL, USA). As the χ^2^ goodness of fit statistic and the Goodness of Fit Index (GFI) are very sensitive to sample size, and given that our sample is large (*N* = 1187) indeed, we present numerous alternative goodness of fit indices (Tucker Lewis Index (TLI), Comparative Fit Index (CFI), and Root Mean Square Error of Approximation (RMSEA)) in this contribution [80]. It is generally suggested that the TLI and CFI should exceed 0.90, or even 0.95, for the model to be considered a good fit. Similarly, RMSEA should be lower than 0.08, better is 0.05, to reflect a good fit [81]. Additionally, the joint significance test as recommended by MacKinnon [82,83] was used to investigate whether the hypothesized mediation effects exist. The two conditions that must be met to conclude that a mediating effect exists are as follows: (1) the independent variable is significantly related to the mediating variable; and (2) the mediating variable is significantly related to the dependent variable. The significance of the mediated effect of the specific independent variable on the dependent variable was calculated using Sobel’s [84] test. Our outcomes indicate that the two conditions as stated above have been met, and Sobel’s [84] test for mediation showed that both mediation effects were significant. 

## 4. Results

### 4.1. Preliminary Analyses

Table 1 presents the means, standard deviations, and inter-correlations among all study variables. Leadership quality and supervisory social support were rather highly correlated (0.63), but did not exceed the value that would pose a serious threat to the model [85]. All reliability measures (Cronbach’s alpha) are in the good to excellent range, that is, 0.70 to 0.99 [86].

Examination of the job demands shows that nurses reported higher than average levels of emotional job demands (M = 3.45; SD = 0.58), average levels of quantitative job demands (M = 2.99; SD = 0.55), rather high levels of physical demands (M = 3.19; SD = 2.55), and a lower than average level of family-work conflict (M = 1.51; SD = 0.60). The reported levels of job resources by these nurses showed that quality of leadership was slightly higher than average (M = 3.09; SD = 0.76), developmental opportunities were higher than average (M = 3.72; SD = 0.66), social support from supervisors was slightly lower than average (M = 3.04; SD = 0.86), and social support from near colleagues was slightly higher than average (M = 3.71; SD = 0.63). The average level of perceived effort or stress reported was 1.89 (SD = 0.49), the average level of meaning of work reported was 4.22 (SD = 0.57), the average level of burnout was 1.64 (SD = 0.55), and the nurses’ mean intention for occupational turnover was 1.43 (SD = 0.7).

In regard to the control variables, female gender had a significant and positive impact on perceived effort in the under the age of 40 group and was non-significant in the 40 years and over age group (under 40, *β* = 0.09, *p* ≤ 0.01; 40 and over, *β* = 0.04, *NS*); female gender was non-significant in the under the age of 40 group and had a significant and positive impact on meaning of work in the 40 years and over group (under 40, *β* = 0.04, *NS*; 40 and over, *β* = 0.10, *p* ≤ 0.01). Professional tenure had a significant and negative impact on the meaning of work for both age groups; (under 40, *β* = −0.11, *p* ≤ 0.001; 40 and over, *β* = −0.11, *p* ≤ 0.001), while female gender had a significant and positive impact on meaning of work for the 40 plus group only (under 40, *β* = 0.04, *NS;* 40 and over *β* = 0.10, *p* ≤ 0.01). Perceived effort (under 40, *β* = 0.30, *p* ≤ 0.001; 40 and over, *β* = 0.31, *p* ≤ 0.001), and meaning of work (under 40, *β* = −0.08, *p* ≤ 0.05; 40 and over, *β* = −0.04, *p* ≤ 0.05), had a significant impact, in the direction predicted. Family-work conflict appeared to have a direct effect on burnout as well, in addition to its indirect effect; that is the effect that was mediated by the meaning of work. All of these results are based on Time 1 data. Finally, burnout (under 40, *β* = 0.14, *p* ≤ 0.001; 40 and over, *β* = 0.17, *p* ≤ 0.001), had a significant impact on occupational turnover intention in Time 2. With these outcomes, we found preliminary support for our overall hypothesized model.

With the indirect standardized effects of *job demands* included in our research model, we found that the indirect effect of emotional demands, through perceived effort and burnout, on turnover intention was 0.02 (*p* < 0.05) for those nurses under the age of 40 group and 0.021 (*p* < 0.05) for those 40 years and over; the indirect effect of quantitative demands, through perceived effort and burnout, on turnover intention was 0.003 (*p* < 0.001) for those under the age of 40 group and 0.005 (*p* < 0.001) for those 40 years and over; for physical demands, through perceived effort and burnout, on turnover intention was 0.003 (*NS*) for those under the age of 40 group and 0.004 (*p* < 0.05) for those 40 years and over; and for family-work conflict, through perceived effort and burnout, on turnover intention was 0.023 (*NS*) for those under the age of 40 group and 0.033 (*p* < 0.001) for those 40 years and over.

Given the fact that these are significant indirect effects when the structural parameters are constrained to be equal (initial model), with the exception of physical demand and family work conflict for the nurses under the age of 40 group, we concluded that we have found partial support for the assumption that perceived effort and burnout indeed mediate the relationship between the distinguished job demand variables and occupational turnover intention (Hypothesis 1).

In regard to the *job resources* included in our research model, we found that the indirect standardized effect of quality of leadership, through meaning of work and burnout, on turnover intention was 0.0003 (*NS*) for those nurses under the age of 40 group and −0.001 (*p* < 0.05) for those 40 years and over; for developmental opportunities, through meaning of work and burnout, on turnover intention was −0.005 (*p* < 0.001) for those under the age of 40 group and −0.002 (*p* < 0.001) for those 40 years and over; for social support from supervisor, through meaning of work and burnout, on turnover intention was −0.001 (*p* < 0.05) for those under the age of 40 group and −0.001 (*NS*) for those 40 years and over; and for social support from near colleagues, through meaning of work and burnout, on turnover intention was −0.001 (*p* < 0.001) for those under the age of 40 group and −0.0005 (*NS*) for those 40 years and over. 

Given that these were all significant indirect effects when the structural parameters are constrained to be equal (initial model), we concluded that meaning of work and burnout indeed mediate the relationship between the distinguished job resource variables and occupational turnover intention (Hypothesis 2). 

The indirect effect of perceived effort, through burnout, on turnover intention was 4.83 (*p* < 0.001) and for meaning of work, through burnout, on turnover intention was −2.21 (*p* < 0.05). Therefore, we can conclude that burnout indeed mediates the relationship between perceived effort and meaning of work on turnover intention (Hypotheses 3 and 4).

There are significant differences in the means for some of the determinants of occupational turnover intention between those under the age of 40 and those aged 40 years and over. Table 2 presents the means of the determinant variables by age group; those differences that are significant will be discussed here. Emotional demands are greater for those under the age of 40 (3.50 vs 3.40; *p* < 0.001) as are physical demands (30.22 vs 20.94; *p* < 0.001) and family work conflict (1.56 vs 1.46; *p* < 0.01). Developmental opportunities are scored higher for those under the age of 40 (3.82 vs 3.63; *p* < 0.001) as is social support from colleagues (3.84 vs 3.60; *p* < 001). Those under the age of 40 reported higher perceived effort (11.53 vs 11.02; *p* < 0.01). Those 40 and over report higher level of burnout (1.71 vs 1.57; *p* < 0.001). Not surprisingly, professional tenure is greater for those in the 40 and over age group, (19.10 vs 7.94; *p* < 0.001). Table 3 presents the distribution of occupational turnover intention for both age groups. The distribution appears to be significantly different between the two groups; F = 6.536; *df* 1, 1185; *p* < 0.01, herewith confirming Hypothesis 5.

### 4.2. Model Fit and Hypotheses’ Tests

In order to test Hypothesis 6 regarding the moderating effects of age, we conducted multi-group Structural Equation Modelling (SEM) analysis in AMOS as follows.

Step 1: estimated the unconstrained model where all structural paths were allowed to be different for the two age groups.

Step 2: compared the fit of the unconstrained model with the fit of the model that constrained all structural relationships to be equivalent.

The outcomes of our hypotheses’ tests indicate that, without constraining any of the structural paths when estimating the parameters, provided a satisfactory fit to the data, χ^2^ = 232.09, *df* = 108, CFI = 0.95, RMSEA = 0.030, IFI = 0.95, TLI = 0.92; see Table 4. Additionally, Table 4 shows that with each additional constraint applied to the model, the fit of the model deteriorates significantly. Constraining the structural paths in Step 2 resulted in χ^2^ = 13.82, df = 108, CFI = 0.95, RMSEA = 0.031, IFI = 0.95, TLI = 0.91; these differences are significant at *p* < 0.001. This deterioration of results indicates that the best fit will be required if to apply no constraints to the model; i.e., standardized estimates separately for each age group (see Figure 1).

Therefore, we report the estimates of the two age groups separately, as seen in Table 5. Numerous determinants of occupational turnover intention differ significantly as regards their impact for nurses under 40 and for nurses from the 40 and over age group.

As Table 5 shows, with a few exceptions, the path coefficients were significant at a minimum of *p* ≤ 0.05, herewith supporting our overall hypothesized model. The mediating effects of burnout, in the relationships between perceived effort and work meaning, respectively, as the determinants, and with occupational turnover intention as the outcome variable, were significant; for perceived effort at *p* < 0.001 and for work meaning at *p* < 0.05.

The determinants of perceived effort that were similar for both age groups are emotional demands (under 40, *β* = 0.08, *p* ≤ 0.05; 40 and over, *β* = 0.09, *p* ≤ 0.05) and quantitative demands (under 40, *β* = 0.49, *p* ≤ 0.001; 40 and over, *β* = 0.44, *p* ≤ 0.001). The remaining determinants were dissimilar in terms of whether or not they are significant; physical demands (under 40, *β* = 0.06, *non-significant (NS)*; 40 and over, *β* = 0.09, *p* ≤ 0.05), and family-work conflict (under 40, *β* = 0.05, *NS*; 40 and over, *β* = 0.14, *p* ≤ 0.001), had a significant impact on perceived effort. 

The determinant of meaning of work that was similar between the two groups comprises development opportunities (under 40, *β* = 0.43, *p* ≤ 0.001; 40 and over, *β* = 0.37, *p* ≤ 0.001). The remaining determinants of meaning of work were dissimilar; leadership quality (under 40, *β* = 0.03, *NS*; 40 and over, *β* = 0.10, *p* ≤ 0.05), social support from one’s supervisor (under 40, *β* = 0.08, *p* ≤ 0.05; 40 and over, *β* = 0.06, *NS*), and social support from near colleagues (under 40, *β* = 0.13, *p* ≤ 0.001; 40 and over, *β* = 0.05, *NS*). 

Significant differences were observed between the nurses under 40 years old and the 40 and over age groups in each stage of our model; the standardized estimates are used in all of the following discussion of the specific results. These results clearly indicate differences between the under 40 and 40 and over age groups in terms of the importance of factors determining occupational turnover intention; herewith supporting Hypothesis 6. The most striking differences are in those variables that are significant determinants for one of the two distinguished age groups and non-significant for the other age group (refer to Table 5). For example, physical demands and family-work conflict are significant in determining perceived effort for those 40 and over, but have no impact for those nurses under 40, while gender (female) has a significant, positive impact on perceived effort, yet only for those under the age of 40. Meaning of work is determined by developmental opportunities, social support from supervisor and social from colleagues for those under 40, but only by leadership quality and developmental opportunities for those aged 40 and over.

## 5. Discussion

The most important findings in this study can be summarized as follows. In line with our expectations, burnout symptoms appear to be predicted by perceived effort, which significantly increased burnout, while work meaning significantly decreased burnout. Nurses’ turnover intentions were predicted by burnout symptoms; an increase in burnout resulted in a significant increase in the intention to leave the nursing profession. In particular, the impact of perceived effort and meaning of work on burnout are not equivalent, and in opposite directions. 

As perceived effort is significantly predicted by nurses’ job demands, while meaning of work is predicted by their available job resources, it is important for health care management to carefully consider the possible impact of these factors at the workplace. That is to say, from the specific outcomes of our study, we suggest that while increasing job resources may be effective to protect nurses’ well-being, the far greater impact would result from decreasing those job demands that increase perceived effort. A closer examination of the impact of the job demands on perceived effort reveals that quantitative demands have a far greater impact on perceived effort than any other job demand included in our study. Examination of the standardized coefficients shows outcomes of 0.08, 0.09 (under 40, 40 and over) for emotional demands, 0.49, 0.44 for quantitative demands, 0.06, 0.09 for physical demands, and 0.05, 0.14 for family-work conflict. These outcomes indicate that quantitative demands have an impact that is approximately four times higher than any other of the job demands on perceived effort.

As far as the investigated job resources are concerned, we have found that developmental opportunities (0.43, 0.37) had a far greater impact than the other job resources, followed by social support from one’s colleagues (0.13, 0.08). The remaining coefficients show outcomes of 0.03, 0.10 for quality of leadership, 0.43, 0.37 for developmental opportunities, 0.08, 0.06 for social support from one’s supervisor, and 0.13, 0.08 for social support from one’s colleagues. Developmental opportunities had an impact being four times higher than any other of the job resources on perceived meaning of work. 

The outcomes of our study shed more light on possible measures health care management can take to prevent occupational turnover. The majority of previous research in this scholarly field has focused on job turnover [12,87], while leaving the profession completely is a much more serious threat for societies and countries given the negative impact on the overall supply of nurses [1,88,89]. Our research provides valuable empirical insight into important reasons for leaving the nursing profession. Specifically, we have shown that, on the one hand, quantitative demands increase perceived effort the most, while, on the other hand, developmental opportunities increase work meaning the most. In turn, perceived effort in particular and work meaning, albeit it to a lesser extent, are associated with burnout levels, respectively in a positive and a negative way. Our results suggest that the greatest impact in terms of preventing occupational turnover intention may come from efforts from management and other stakeholders in health care institutions that are directed explicitly to reduce the quantitative demands on nurses.

Additionally, our outcomes demonstrate that it is necessary to group nurses by age category to obtain accurate and generalizable results regarding the determinants of occupational turnover intention. These are necessary in creating meaningful management interventions.

### Limitations of this Study and Recommendations for Future Research

As we have used self-report measures for all model variables, a common-method bias might exist [90]. In order to increase the validity of the outcomes, nurses’ self-assessments and supervisor assessments might be combined in future research. Another limitation of our study is that the results should be viewed in light of the data having been collected in the health care industry only, and from one profession, i.e., nursing. This may cast some doubt on the suitability of generalization to other professions or industry sectors. Nevertheless, as our results are in line with the theory and the pattern of relationships as assumed, we think they are noteworthy and provide challenges for future research. 

Moreover, we have focused on nurses’ intention to leave the profession instead of actual turnover behavior. There are theoretical and practical reasons for studying intention rather than behavior. Previous turnover research [91,92,93] reported that turnover intention is a stronger predictor of actual turnover than other variables [94]. Furthermore, using intention to leave the profession as an indicator overcomes the fact that actual turnover is a low base rate event. For organizations, occupational turnover intention may be interpreted to be a highly useful variable, even more so than actual leave. After all, if health care organizations are aware of a high prevalence of occupational turnover intention, they may still take action in order to retain the nurses. Still, future research is needed to establish the predictive validity of our overall hypothesized model for actual occupational turnover. 

## 6. Conclusions

From an individual, organizational, and social perspective; there is a critical need to better understand why so many nurses develop an intention to leave their profession. Our findings reveal that the largest decrease in burnout, and the resulting occupational turnover intention, will be obtained by diminishing nurses’ job demands and increasing their job resources. Head nurses have a major responsibility to protect nurses’ employability; they should, on a daily basis, provide high-quality leadership, safeguard ample opportunities for career development, and provide strong social support to cope with all stressors at the workplace. Unfortunately, head nurses’ leadership quality can vary substantially; many who are promoted to the position of head nurse are not carefully screened regarding their leadership competencies and previous experience in managing people. Therefore, it is imperative that line management in health care organizations have sufficient training that enables them to discuss important HRM issues with colleagues who have specific expertise in this field.

Managers in health care settings that do not provide satisfactory job resources and other forms of (career) support to help nursing employees cope with ever-increasing job demands, and that fail to determine their lack of resources—will experience growing levels of burnout among their staff, which may result in premature departure. If the lack of resources is only slight, job satisfaction and morale are reduced. A more serious lack of job resources will result in increased turnover intentions, due to increased levels of burnout. Moreover, it is important for health care institutions to prioritize finding ways to increase the opportunities to obtain social support for all staff members. Social support could be improved, for example, by creating social networks. In addition, head nurses can develop an atmosphere in which staff members are encouraged to identify stress factors within the work environment, and wherein it is possible to learn from mistakes.

Employees working in nursing roles are exposed to emotional involvement, stress, work constraints, and role uncertainty, making the need to talk things through with colleagues and supervisors an important job resource. When it comes to situations of psychological stress, colleagues appear to be the most important source of support, particularly when institutionally that kind of support is lacking [95]. Hospitals and other health care organizations that employ nurses are not without options to proactively address increased nurse turnover. Our findings show that the organizational or management interventions that will have the greatest impact in preventing increased turnover are two-fold: one should reduce the quantitative demands on nurses and one should increase the developmental opportunities available to provide them support. These two findings apply to both younger and older nurses, so implementing management interventions for them should be prioritized.

## Figures and Tables

**Figure 1 ijerph-16-02011-f001:**
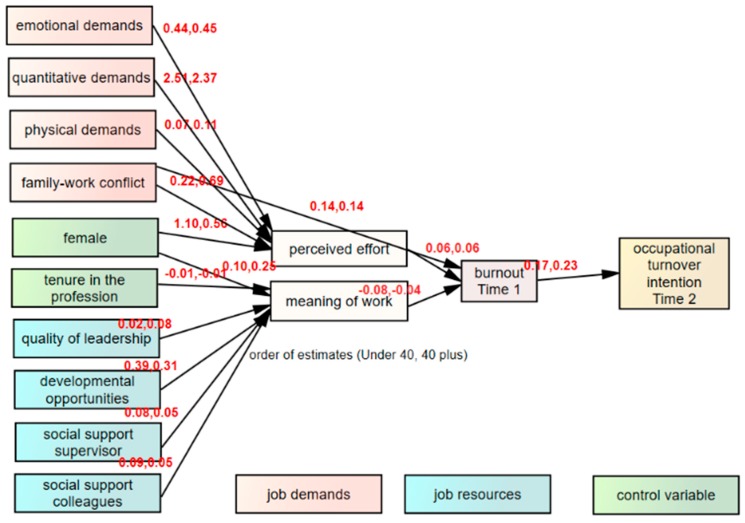
Nursing Sector-Specific Model on Occupational Turnover Intention for younger versus older nurses; standardized estimates.

**Table 1 ijerph-16-02011-t001:** Means, Standard Deviations, Reliability Coefficients (Cronbach’s alpha; on the diagonal), and Correlations Between the Model Variables, *N* = 1187.

	Variable	M	SD	1	2	3	4	5	6	7	8	9	10	11	12	13	14	15
1	female	94.44	0.23	-														
2	Tenure in the profession	13.62	8.57	0.01														
3	age	39.80	9.68	0.02	0.74 **													
4	Emotional demands	3.45	0.58	−0.15 **	0.02	−0.01 **	*0.70*											
5	Quantitative demands	2.99	0.55	0.55	0.03	−0.06 *	0.28 **	*0.70*										
6	Physical demands	3.19	2.55	0.02	−0.16 **	−0.30 **	0.27 **	0.31 **	*0.87*									
7	Family-work conflict	1.51	0.60	−0.04	−0.05	−0.11 **	0.03	0.14 **	0.06	*0.85*								
8	Quality of leadership	3.09	0.76	0.05	−0.05	−0.03	−0.01	−0.15 **	−0.05	−0.08 **	*0.87*							
9	Developmental opportunities	3.72	0.66	−0.04	−0.07 *	−0.19 **	0.21 **	0.12 **	0.08 **	−0.01	0.22 **	*0.75*						
10	Social support; supervisor	3.04	0.86	0.00	−0.06 *	−0.04	0.00	−0.12 **	−0.00	−0.05	0.63 **	0.20 **	*0.84*					
11	Social support; colleagues	3.71	0.63	−0.01	−0.13 **	−0.27 **	0.14 **	0.06 *	0.16 **	−0.02	0.13 **	0.29 **	0.20 **	*0.77*				
12	Perceived effort	1.89	0.49	−0.07 *	-0.01	−0.11 **	0.23 **	0.53 **	0.26 **	0.16 **	−0.12 **	0.12 **	0.11 **	0.03	*0.85*			
13	Meaning of work	4.22	0.57	0.06	−0.11 **	−0.10 **	0.06 *	0.04	0.05	−0.05	0.24 **	0.45 **	0.24 **	0.21 **	0.00	*0.81*		
14	Burnout	1.64	0.55	0.00	−0.08 **	−0.15 **	0.11 **	0.23 **	0.09 **	0.21 **	0.11 **	0.02	−0.07 *	0.00	0.34 **	−0.63 **	*0.83*	
15	Occupational turnover	1.43	0.70	−0.08 *	−0.01	−0.06 **	0.02	0.06 *	−0.00	0.10 **	0.14 **	−0.09 **	−0.10 **	0.02	0.06 **	−0.19 **	−0.16 **	*0.85*

Note. a. * *p* < 0.05, ** *p* < 0.01, b. Means and standard deviations of binary (0,1) coding for supervisor-subordinate age difference, c. Binary coding for gender (1,2).

**Table 2 ijerph-16-02011-t002:** Means and Significant Differences by Age Group.

Determinants of Occupational Turnover Intention	Age < 40	Age > 40
Emotional demands ***	3.50	3.40
Quantitative demands	3.00	2.97
Physical demands ***	30.22	20.94
Family-work conflict **	1.56	1.46
Quality of leadership	3.10	3.07
Developmental opportunities ***	3.82	3.63
Social support, from supervisor	3.08	3.00
Social support, from colleagues **	3.84	3.60
Perceived effort **	11.53	11.02
Meaning of work	4.25	4.19
Burnout ***	1.57	1.71
Gender	94.00%	94.87%
Professional tenure ***	7.94	19.10
Occupational turnover intention	1.46	1.40

** *p* < 0.01, *** *p* < 0.001.

**Table 3 ijerph-16-02011-t003:** Occupational Turnover Intention Distribution.

How often have you thought about giving up nursing completely?	Less than 40; Age in years	40 and over; Age in years
Never	63.45%	69.38%
Several times per year	29.14%	23.88%
Several times per month	5.69%	4.15%
Several times per week	1.38%	2.08%
Every day	0.34%	0.52%

F = 6.536; *df* 1, 1185; *p* < 0.01.

**Table 4 ijerph-16-02011-t004:** Goodness-of-fit Indices for Alternative Models.

Model	χ^2^	*df*	CFI	RMSEA	IFI	TLI	Δχ^2^	Δ *df*
Unconstrained	232.09	108	0.95	0.030	0.95	0.92		
Structural weights	245.91	123	0.95	0.031	0.95	0.91	13.82 ***	15
Structural intercepts	272.87	127	0.94	0.031	0.94	0.91	40.78 ***	19
Structural means	965.61	137	0.64	0.071	0.64	0.52	733.52 ***	29
Structural covariances	1199.94	169	0.55	0.072	0.55	0.51	967.85 ***	61
Structural residuals	1205.27	173	0.55	0.071	0.55	0.52	973.17 ***	65

*** *p* < 0.001.

**Table 5 ijerph-16-02011-t005:** Estimated Regression Coefficients from the Structural Model for Each Age Group (standardized coefficients in brackets).

Determinants of Occupational Turnover Intention	Perceived Effort		Meaning of Work		Burnout		Occupational Turnover Intention	
	Under 40	40 plus	Under 40	40 plus	Under 40	40 plus	Under 40	40 plus
Emotional demands	0.44 (0.08) *	0.45 (0.09) *						
Quantitative demands	2.51 (0.49) ***	2.37 (0.44) ***						
Physical demands	0.07 (0.06)	0.11 (0.09) *						
Family-work conflict	0.22 (0.05)	0.69 (0.14) ***			0.14 (0.15) ***	0.14 (0.15) ***		
Quality of leadership			0.02 (0.03)	0.08 (0.10) *				
Developmental opportunities			0.39 (0.43) ***	0.31 (0.37) ***				
Social support from supervisor			0.08 (0.08) *	0.05 (0.06)				
Social support from colleagues			0.09 (0.13) ***	0.05 (0.08)				
Perceived effort					0.06 (0.30) ***	0.06 (0.31) ***		
Meaning of work					−0.08 (−0.08) *	−0.04 (−0.04) *		
Burnout							0.17 (0.14) ***	0.23 (0.17) ***
Gender	1.10 (0.09) **	0.56 (0.04)	0.10 (0.04)	0.25 (0.10) **				
Professional tenure			−0.01 (−0.11) ***	−0.01 (−0.11) ***				

* *p* < 0.05, ** *p* < 0.01, *** *p* < 0.001.

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
