# Peer review of "Impact of Job Demands and Resources on Nurses’ Burnout and Occupational Turnover Intention Towards an Age-Moderated Mediation Model for the Nursing Profession"

_ijerph, 2019, doi:10.3390/ijerph16112011_

Round 1

Reviewer 1 Report

·         Summary of the manuscript

The purpose of this manuscript is to investigate the impact of job demands and resources on nurses’ burnout and occupational turnover intention by adopting and generalizing job demand-resource framework. It found that sub-dimensions of job demand have impacts on burnout with the mediation of perceived effort/strain and those of job resources buffer the burnout with the mediation of meaning of work. It also found that the burnout in time 1 leads to occupational turnover intention in time 2 (after 12 months). The manuscript is well-written in general and contains useful insights.

·         General comments

- Although the manuscript is well written, the reviewer wonders if the authors would submit it right after being rejected in other journal. It does NOT follow the formatting rules of MDPI journal. The reviewer believes that the authors could have shown more respect to the journal and reviewer by following the rules.

- Abstract is too long to understand what the authors want to say at a glance. Also, the abstract is not matched to the formatting rule of MDPI journal. Considering the purpose and format of the abstract, the reviewer strongly recommend to rewrite and shorten the abstract.

- The authors must follow the formatting rules for in-text citations and bibliography. Both the in-text citations and the bibliography style are not written in the MDPI style.

- As though Introduction section is well-written in general, the reviewer believes that it may enrich the contribution of the research to address the finding and implications in the section. Now that the summary for findings and implications is totally absent, the reviewer feels like that Induction section may end with unclear purposes, methodology, and results.

- Also, Theory section must be rewritten for the readers to follow which hypothesis is driven from which theoretical background. Since reviewing the previous literature was conducted in a highly limited way, following the development such hypotheses from the theoretical background can be a hard time for the readers.

- In Methods section, what does it mean by
“In total, 272 health care institutions (nine hospitals, thirteen nursing and old peoples’ homes, and five home care institutions) decided to participate in our study.”
from line 240 to 242?
The number of health care institutions participating in the study does NOT match to the sum of numbers in parenthesis.

- In Analysis section, Table 2 and Table 3 are written by classifying the nurses into two groups by splitting them either less than 40 or 40 and over. Although the mean of the nurses’ ages is very close to 40, the standard deviation is relatively large (9.68), which means that the number of nurses aging between 30.12 and 49.48 takes 2/3 of the target nurses. Therefore, we have to choose the classification criteria in a very caution way. However, ‘40’ is lack of persuasiveness.

- Figure 1 has no caption. The research model does not match with the hypotheses, Also, please rewrite the figure with the factor loadings.

Author Response

Hereby, we will reply point-by-point to all comments/feedback that has been raised. Upon request by the editorial office, we have highlighted all changes in the Revised Manuscript using the track changes function.

Reviewer 1:

Summary of the manuscript

The purpose of this manuscript is to investigate the impact of job demands and resources on nurses’ burnout and occupational turnover intention by adopting and generalizing job demand-resource framework. It found that sub-dimensions of job demand have impacts on burnout with the mediation of perceived effort/strain and those of job resources buffer the burnout with the mediation of meaning of work. It also found that the burnout in time 1 leads to occupational turnover intention in time 2 (after 12 months). The manuscript is well-written in general and contains useful insights.

Dear Reviewer, thank you very much for your positive feedback on our manuscript and for the compliments on its writing style. We are glad that you appreciate its insights and will do our utmost to respond to you below how we have dealt with your constructive feedback.

·         General comments

- Although the manuscript is well written, the reviewer wonders if the authors would submit it right after being rejected in other journal. It does NOT follow the formatting rules of MDPI journal. The reviewer believes that the authors could have shown more respect to the journal and reviewer by following the rules.

Dear Reviewer, the manuscript has been carefully aligned with the author guidelines of IJERPH, except for the references indeed. Given the fact that we wanted to give the Special Issue Editor Dr. Nygård more time to consider the suitability of this manuscript for the special issue, we have consulted the editorial assistant before submission. She has given us the opportunity to submit the article using APA referencing style for the first round, as that is the first author’s common expertise. In this revision round, we have used the expertise of the second author who has a medical background. The second author has conscientiously converted all references after this thorough revision round using the formatting rules of IJERPH.

- Abstract is too long to understand what the authors want to say at a glance. Also, the abstract is not matched to the formatting rule of MDPI journal. Considering the purpose and format of the abstract, the reviewer strongly recommend to rewrite and shorten the abstract.

Dear Reviewer, the abstract has been written conscientiously, and we have double-checked the formatting rules of IJERPH. The length of the abstract (231 words now) and the number of key words (8 keywords) are in line with the author guidelines.

- The authors must follow the formatting rules for in-text citations and bibliography. Both the in-text citations and the bibliography style are not written in the MDPI style.

Dear Reviewer, we apologize, yet please also see the reason for this as explained above. In this revision round, we have used the expertise of the second author who has a medical background. The second author has conscientiously converted all references after this thorough revision round using the formatting rules of IJERPH.

- As though Introduction section is well-written in general, the reviewer believes that it may enrich the contribution of the research to address the finding and implications in the section. Now that the summary for findings and implications is totally absent, the reviewer feels like that Induction section may end with unclear purposes, methodology, and results.

Dear Reviewer, thank you very much for your compliments and for your insightful feedback. Following your request, we have now added the findings and implications into this section. Please, see all track changes in the Revised Manuscript’s Introduction section.

- Also, Theory section must be rewritten for the readers to follow which hypothesis is driven from which theoretical background. Since reviewing the previous literature was conducted in a highly limited way, following the development such hypotheses from the theoretical background can be a hard time for the readers.

We have tried our best to clearly explain the underlying theoretical reasoning for the distinguished hypotheses, and have added more references. We hope that by restructuring the Theoretical framework, and by clearly aligning the hypotheses with the specific lines of argumentation on which they are based, to have added more focus into the manuscript.

- In Methods section, what does it mean by
“In total, 272 health care institutions (nine hospitals, thirteen nursing and old peoples’ homes, and five home care institutions) decided to participate in our study.”
from line 240 to 242?

The number of health care institutions participating in the study does NOT match the sum of numbers in parenthesis.

Dear Reviewer, sorry about this typo, and thank you for your careful review of our manuscript. This typo has now been corrected; 27 health care institutions participated in our study.

- In Analysis section, Table 2 and Table 3 are written by classifying the nurses into two groups by splitting them either less than 40 or 40 and over. Although the mean of the nurses’ ages is very close to 40, the standard deviation is relatively large (9.68), which means that the number of nurses aging between 30.12 and 49.48 takes 2/3 of the target nurses. Therefore, we have to choose the classification criteria in a very caution way. However, ‘40’ is lack of persuasiveness.

Dear Reviewer, building upon ample previous work by Van der Heijden and colleagues (see for instance, Van der Heijden et al., 2009), [(see also Finkelstein & Farrell, 2007, p. 100 on the Age Discrimination in Employment act (ADEA)], we categorize nurses into younger (under 40 years) versus older (≥ 40 years old) ones. In order to further justify our age-related hypotheses, we have elaborated on the underlying theorizing and have thoroughly rewritten the accompanying section in the Theoretical framework.

- Figure 1 has no caption. The research model does not match with the hypotheses, Also, please rewrite the figure with the factor loadings.

Dear Reviewer, Figure 1’s Caption was added in the previous version of the manuscript and was entitled ‘Hypothesized Research Model’.

Figure 1’s caption has now been changed into ‘Figure 1. Nursing Sector-Specific Model on Occupational Turnover Intention for younger versus older nurses; standardized estimates.

Moreover, the estimates have been incorporated into the Figure.

Reviewer 2 Report

Paper is confusing with too many topics. Pick at the most two hypotheses, not six.  A separate paper based on fewer topics can be organized into a minimum of three or maybe four articles; don't try to cover everything in one article. What you have can be the resource for a book, separated into chapters based on topics. Limiting topics will make article more interesting and less confusing; for example, either turnover, burnout and stress, leadership, resources, or age.  Pick one.  Also, abstract is too long.  300-350 words is the normal or common length of an abstract.  Some of content should be in body of paper, e.g., methodology. Purpose of research, population, and findings will work of the abstract. Leave the stats out of the abstract.

Author Response

Hereby, we will reply point-by-point to all comments/feedback that has been raised. Upon request by the editorial office, we have highlighted all changes in the Revised Manuscript using the track changes function.

Reviewer 2:

Paper is confusing with too many topics. Pick at the most two hypotheses, not six.  A separate paper based on fewer topics can be organized into a minimum of three or maybe four articles; don't try to cover everything in one article. What you have can be the resource for a book, separated into chapters based on topics. Limiting topics will make article more interesting and less confusing; for example, either turnover, burnout and stress, leadership, resources, or age.  Pick one.  Also, abstract is too long.  300-350 words is the normal or common length of an abstract.  Some of content should be in body of paper, e.g., methodology. Purpose of research, population, and findings will work of the abstract. Leave the stats out of the abstract.

Dear Reviewer, thank you very much for your constructive and insightful feedback which has helped us a lot in adding more focus to the manuscript.

We have tried our best to clearly explain the underlying theoretical reasoning for the distinguished hypotheses, and have added more references. We hope that by restructuring the Theoretical framework, and by clearly aligning the hypotheses with the specific lines of argumentation on which they are based, to have added more focus into the manuscript.

In addition, following up on an earlier request from the Special Issue Editor, Dr. Nygård, some more explicit hypotheses on the role of age have been added. After a thorough consultation with a statistician, being the second author, we do believe that empirically investigating our overarching mediation model entails the need to conscientiously test all of its underlying hypotheses, and to discuss all of their outcomes (see also the comments of Reviewer 1 in this regard). Please, let us know in case you have any remaining concerns.

In addition, the abstract has been written conscientiously, and we have double-checked the formatting rules of IJERPH. The length of the abstract (231 words now) and the number of key words (8 keywords) are in line with the author guidelines.

To conclude, we have also performed another proofreading, even while the second author is native American speaker, so have done our best to meet your expectations in terms of the use of English language, and to eliminate all possible flaws in this regard.

Round  2

Reviewer 1 Report

The review really appreciates the authors’ effort to reflect all of the comments into the revised manuscript and to response them one-by-one. The revised version was totally improved to follow MDPI formats in Abstract, in-text citations, and bibliography.

The main concern of mine for the original version was the theoretical part and hypotheses development. I believe that the authors exert their own best to revise that part.

However, it still looks awkward to me that 4 hypotheses are derived in the end of a paragraph at the same time. Please improve the theoretical part into a more systematic way.

At least in current form, the development process of such hypotheses does NOT look clear and straightforward to me.

Author Response

Dear Reviewer,  First of all, we would like to thank you very much again for your timely feedback and your guidance along this review process. We are happy that you appreciate the care we took with our review and that our response letter is so positively evaluated. It is good to see that our efforts are so well understood. In response to your last review report:  As this contribution comprises the first empirical work that uses a Structural Equation Model wherein simultaneous testing is used to investigate a model with perceived effort (or stress), being predicted by job demands, and meaning of work, being predicted by job resources, simultaneously impacting burnout; and with burnout mediating their relationship with occupational turnover intention of nurses, we have followed APA style in our theoretical argumentation as well. In particular, the first four hypotheses are build upon the Job Demands-Resources model and we have incorporated both a health-impairment process and a motivational process line of reasoning into our, therefore, integrative theoretical framework. Next, we have worked towards a sound argumentation for our Multi-group Age moderation hypotheses, which is written in another section (as common in APA articles), resulting in the last two hypotheses. If we would break up the article in different sections for each hypothis, we would have to repeat a lot of the theorizing using the JD-R model.   We hope that you are with us in that this was the optimal approach, which is moreover in line with scholarly conventions.  Moreover, we would like to stress again that our second author is native American speaker and that writing has been carefully checked again in this latest version.